# Inspection of antimicrobial remains in bovine milk in Egypt and Saudi Arabia employing a bacteriological test kit and HPLC-MS/MS with estimation of risk to human health

**Nora H. Al-Shaalan**[1][☉], **Jenny Jeehan Nasr**[2][☉], **Shereen Shalan**[2][☉], **Areej M. El-Mahdy**[3,4][☉]*

**1** Chemistry Department, College of Science, Princess Nourah Bint Abdulrahman University, Riyadh, Saudi Arabia, **2** Department of Pharmaceutical Analytical Chemistry, Faculty of Pharmacy, Mansoura University, Mansoura, Egypt, **3** Department of Pharmaceutical Sciences, College of Pharmacy, Princess Nourah Bint Abdulrahman University, Riyadh, Saudi Arabia, **4** Department of Microbiology and Immunology, Faculty of Pharmacy, Mansoura University, Mansoura, Egypt

☉ These authors contributed equally to this work.
* amalkasaby@pnu.edu.sa

**Data Availability Statement:** All relevant data are within the manuscript.

## Abstract

Veterinary medicine uses antibiotics randomly for treatment and growth promotion. Milk of dairy animals contains substantial quantities of antibiotics that have harmful effects on health. It is therefore necessary to test commercially available milk using immunological, chromatographic, or microbiological methods to confirm the absence of antibiotic residues. This study aims to perform a microbiological test, followed by a quantitative confirmation analysis, on raw milk to assess the presence of antibiotic residues. Tests were conducted on 200 milk samples collected from markets and farms in Saudi Arabia and Egypt. The microbial inhibitor test (Delvotest SP-NT) revealed that 40 samples were positive for antibiotic residues. The positive samples were further tested using liquid chromatography-tandem mass spectrometry (LC-MS/MS) as a confirmatory quantitative test for 29 antibiotics that belong to five groups: tetracyclines, sulfonamides, fluoroquinolones, macrolides, and lactamases. Only four samples tested positive for oxytetracycline residues above the maximum residue limit. Based on these results, researchers suggest a monitoring system that considers both microbial and HPLC-MS/MS methods when detecting antibiotic residues in bovine milk. The analysis of risk to human health revealed that antibiotic residues at the detected levels do not pose any health risks to consumers.

## 1. Introduction

Antibiotics are a broad group of medicines that are utilized to kill or prevent the growth of bacterial microorganisms. In the field of human medicine, they are used to treat a variety of microbial infections. In the field of animal husbandry, they are widely used for infection treatment, but they are also illegally used for prophylaxis and growth promotion, which can boost a

**Funding:** This research was funded by The Deanship of Scientific Research at Princess Nourah bint Abdulrahman University, through the Research Funding Program (Grant No# FRP-1440-26).

**Competing interests:** The authors have declared that no competing interests exist.

farm's financial return [1, 2]. The use of antibiotics in dairy animals may result in presence of antibiotic residues into milk, triggering unfavorable allergic reactions in humans [3]. Moreover, antimicrobials may cause antibiotic-resistant bacteria to flourish, which can cause serious medical conditions [4]. Various governments have established monitoring projects to assess antibiotic concentrations in food and to establish a maximum residue limit (MRL) for them [5, 6]. Consequently, the European Union (EU) has prohibited the illegal usage of antibiotics for growth promotion. Global organizations and State governing agencies like the World Health Organization (WHO) and the United Nations Food and Agriculture Organization (FAO) (collectively formed the Codex Alimentarius), and the United States Department of Agriculture (USDA) have established MRLs for medicines intended in veterinary usage and their presence is allowed in nutrients of animal source [7]. In developing countries, however, veterinary drug abuse is detected at shocking rates due to inadequate supervision and limited analytical controls [8].

Checking nutrients of animal sources for the existence of antibiotic residues is generally executed by screening procedures, which comprise microbiological tests, and confirmatory quantitative procedures including liquid chromatography coupled to mass spectrometry [9]. Microbiological tests consist of two major groups: tube and multi-plate tests. The multi-plate test utilizes of dishes holding agar medium with diverse indicator bacteria. Specimens are placed on the top of the agar surface, then, after incubation, the growth of bacteria will change the opacity of the agar. If antibiotic residues are present in the specimen, inhibition of the growth of bacteria will occur, producing a clear zone around the specimen. A major disadvantage of this test is that it is time-consuming due to the continuous need for fresh agar plates and fresh bacterial cultures; hence, they are inappropriate for macroscale [10].

Tube tests, which are commercially accessible, and are employed for onsite screening of antibiotic residues. The tube tests are ready-to-use tubes containing an indicator microorganism together with the nutrients, a pH indicator, and an agar medium [11]. The tube methods are used efficiently for the examination of residual antibiotic drugs in the livestock food and are used more commonly than the multi-plate test as it is less time-consuming and less laborious [12]. The most common indicator bacterium used in these inhibition tests is *Geobacillus stearothermophilus*, because of several reasons such as its low contamination level, tolerance to high temperature (55°C), and shorter incubation time (less than 4 h) compared to other bacteria. Furthermore, it is more susceptible to antibiotics, especially, β-lactams [13].

An example of the tube test is the Delvotest® SP NT (produced by DSM Food Specialties Ltd., The Netherlands), which is a standard diffusion test for the screening of antibacterial substances in dairy milk through inhibiting the growth of *Bacillus stearothermophilus* strain, that is susceptible to several antibiotics and sulfa drugs. During the growth of bacteria, they generate acid, thus, altering the agar pH. The presence of antibiotic traces is simply detected by comparing agar colors. This kit requires only 100 μL of the milk sample and only 3 hours incubation. Thus, Delvotest® SP NT is an appropriate, speedy, simple to use, and low-cost alternative test for the detection of numerous antibiotics in milk products. It is also suitable for the simultaneous analysis of a large number of specimens applying a short and simple process. The previous literature reported several analytical procedures describing screening of antibiotic remains in milk [14, 15]. These practices utilized microbial assays and instrumental analysis. Even though analytical methods as HPLC produce precise data of the concentration of residual antibiotics, They require costly apparatus and skilled investigators. Because of the speed and ease of microbiological assays, they are applicable for the prescreening of potential antibiotics. Nevertheless, false negative or false-positive findings might arise when using microbiological testing.

The potential harmful impacts of antimicrobial residues on human health was determined by computing the risk estimation. Normally, estimation of chemical risk involves four distinct

steps; hazard recognition, hazard description/dose-response estimation, exposure evaluation, and risk depiction. Estimation of chemical risk has two extensively utilized concepts, which are; Hazard quotient (HQ) and risk quotient (RQ). The hazard quotient is utilized for the estimation of health risk whilst the risk quotient is employed in the assessment of environmental risk. This methodology is greatly favored for maintaining food safety to ensure public health [15].

The current study aimed to perform prescreening of raw milk specimens to detect antibiotic residues by microbiological inhibitor test kits followed by confirmatory quantitative analysis by an HPLC/MS-MS technique. Furthermore, antibiotic residues in milk were correlated with risk estimation for human health. In this study, the novelty is the application of a microbiological method followed by confirmation chromatography for screening antibiotic residues in milk samples in the Arab region, including Egypt and Saudi Arabia for the first time, and the assessment of associated health risks.

## 2. Materials and methods

### 2.1. Materials

The antibiotic analytical standards; tetracycline HCl, oxytetracycline HCl, chlortetracycline HCl, doxycycline hyclate, nalidixic acid, norfloxacin, ofloxacin, enrofloxacin, ciprofloxacin, lomefloxacin HCl, erythromycin A, oleandomycin triacetate, Tylosin tartrate, tilmicosin, Josamycin, spiramycin, roxithromycin, lincomycin HCl, clindamycin, trimethoprim, sulfadiazine, sulfadoxine, sulfamethoxazole, sulfathiazole, sulfachlorpyridazine, sulfanilamide, and sulfamethoxazole, were obtained from Sigma–Aldrich (Seelze, Germany). The Arab Company for Gelatin and Pharmaceutical Products (Alexandria, Egypt) provided ampicillin and amoxicillin raw materials. Analytical grade Formic acid, disodium hydrogen phosphate dihydrate, citric acid monohydrate, and trichloroacetic acid were of analytical grade and were purchased from Merck (Darmstadt, Germany). Phosphoric acid was from Riedel-deHaën (Seelze, Germany). Preparation of the McIlvaine buffer was prepared by adding 0.1 M citric acid hydrate to 0.2 M disodium hydrogen phosphate (60:40, v/v). The washing solution was prepared by blending water and methanol (95:5, v/v). Twenty percent trichloroacetic acid solution was made for protein precipitation. Regenerated cellulose membrane filters and syringe filters (Minisart RC25) with pore size 0.45 μm were from Sartorius-Stedim (Goettingen, Germany). The solid-phase extraction columns Chromabond ABC18 (C18) sorbent were purchased from Macherey-Nagel, Düren, Germany.

### 2.2. Milk samples

Two hundred bovine milk samples were collected from different locations from Saudi Arabia and Egypt after approval from Institutional Review Board, Princess Nourah bint Abdulrahman University (IRB Log Number: 21–0296) as shown in (Table 1). Samples collected from Saudi Arabia were 100 samples from various markets in Riyadh. One hundred milk samples from Egypt were collected from 7 different Governorates (Cairo, Giza, Dakahlia, Gharbia, Sharkia, Kafr El Sheikh, and Qalyoubia). All milk samples from Saudi Arabia were pasteurized milk of different brands, while Egyptian milk samples were both raw and pasteurized milk of different brands. All samples were kept at 4˚C and analyzed by microbiological test.

### 2.3. Delvotest SP-NT (microbial inhibitor test)

Delvotest SP-NT is a non-specific microbiological test, performed to identify the presence of antibiotic residues in dairy milk. The principle of the test is agar diffusion, in which the agar test tubes contain a fixed standard count of *Bacillus stearothermophilus* spores, nutrient agar,

**Table 1. Collection sampling plan for local raw and pasteurized milk products.**

| Sample type | Country | Sampling region | No. of samples |
|---|---|---|---|
| **Local raw milk** | Egypt | Cairo Governorate | 10 |
| | | Giza Governorate | 10 |
| | | Dakahlia Governorate | 10 |
| | | Gharbia Governorate | 10 |
| | | Sharkia Governorate | 10 |
| | | Kafr El-Sheikh Governorate | 10 |
| | | Qalyoubia Governorate | 10 |
| **Local pasteurized milk** | Egypt | Juhayna products | 5 |
| | | Lamar Egypt products | 5 |
| | | Dina Farms products | 5 |
| | | Almarai products | 5 |
| | | Beyti products | 5 |
| | | Lactel products | 5 |
| **Local pasteurized milk** | Saudi Arabia | **Almarai products** | |
| | | Full cream milk | 5 |
| | | Low-fat milk | 5 |
| | | Full cream laban | 5 |
| | | Low-fat laban | 5 |
| | | Ayran laban | 5 |
| | | **Alsafi products** | |
| | | Full cream milk | 5 |
| | | Low-fat milk | 5 |
| | | Skimmed milk | 5 |
| | | Full cream laban | 5 |
| | | Low-fat laban | 5 |
| | | **Nadec products** | |
| | | Full cream milk | 5 |
| | | Low-fat milk | 5 |
| | | Full cream laban | 5 |
| | | Low-fat laban | 5 |
| | | **Saudia products** | |
| | | Full cream milk | 5 |
| | | Low-fat milk | 5 |
| | | **Activia products** | |
| | | Full cream milk | 5 |
| | | Low-fat milk | 5 |
| | | Full cream laban | 5 |
| | | Low-fat laban | 5 |
| **Total Samples** | | | **200** |

and bromocresol purple as a pH indicator. Delvotest SP NT was purchased from DSM Food Specialties located in Spain [16]. Each sample was added directly to the agar surface (ampoules), then incubated at 64°C for 3 h. After incubation, a color change from purple to yellow was observed due to a change in pH resulting from microbial metabolism. In the case of fermented milk, samples were first heated for 10 min at 80°C to remove natural inhibitors lysozyme and lactoferrin [17]. Test and data interpretation were carried out based on the

manufacturer's instructions. Test results were interpreted visually as 'negative' (yellow agar) and 'positive' (blue or purple agar).

## 2.4. Equipment

Confirmatory chromatographic analysis was executed on an Agilent Technologies HPLC system 1260 (Agilent Technologies, USA). Detection using mass spectrometry was undertaken employing a triple quadrupole API 4500 (ABSciex, Canada), that operates in the positive electrospray ionization under selected reaction monitoring mode. The mass spectrometer settings used were as follows: dwell-time = 20 ms; resolution Q1 and Q3 = unit; nebulizer gas = 12 psi; curtain gas = 12 psi; collision gas = 8 psi; ion spray voltage = 5500 V; temperature = 400˚C. Controlling the hardware and the data procurement and treatment were accomplished utilizing Analyst 1.6.3 Software (ABSciex, Canada). A vortex shaker from Heidolph (Schwabach, Germany), a TDL-60B Centrifuge (Anke, Taiwan), and BHA-180 T Sonicator (Abbotta Corporation, USA) were employed for the sample preparation and the extraction procedure.

## 2.5. Chromatographic conditions

All conditions used were according to the reference method. LC analyses were executed using a Nucleodur MN-C18 column (150 mm × 4.6 mm i.d., 5 µm particle size), Macherey-Nagel, Düren, Germany. The mobile phase used was a gradient of parts A (water containing 0.2% formic acid) and B (acetonitrile containing 0.2% formic acid) at an oven temperature of 30˚C with a flow rate of 0.3 mL/min. The gradient started with 90% of eluent A for 1 min, then reduced to 40% for 11 min. This composition was kept steady for 3 min, then was raised to 90% of eluent A within 1 min.

## 2.6. Stock standard solutions

Preparation of stock standard solutions of all analytical standards was performed by accurately weighing the materials that were dissolved in methanol. But, for solubilization of quinolones, it is necessary to add 2% of a 2 M ammonium hydroxide solution to methanol solution. All stock solutions of concentration 1 mg/mL were stored in the refrigerator. Working standard mixed solutions of each group of antibiotics were prepared in a concentration of 20-fold MRL for tetracyclines, macrolides, lincosamides, and quinolones and 10-fold MRL for sulfonamides through dilution with the mobile phase (water and acetonitrile 90:10, v/v, with 0.2% formic acid). If no MRL existed, the concentration of the analyte in the mixture was 0.2 µg/mL.

## 2.7. Preparation of samples

Preparation of milk samples was performed adopting a reference method [18], in which, 5 mL of each test milk samples were transferred to a centrifuge tube. The samples were mixed with 100 µL of trichloroacetic acid solution 20% (w/v) and the mixtures were vortexed. 10 mL of McIlvaine buffer at pH 4.0 were added and the mixtures were vortexed for 1 min and then, subjected to centrifugation at 4000 rpm for 15 min. The supernatant was removed and filtered. The filtrate was then exposed to a solid-phase extraction procedure. First, conditioning of the SPE cartridges was made using 6 mL of methanol and 6 mL of water. The filtrate was then moved to the cartridge. The washing step was done with 6 mL of 5% methanol in water (v/v), the cartridges were then dried for 10 min. Then, elution of the analytes was done using 6 mL methanol, followed by vaporization to dryness in nitrogen. The residue was redissolved in 1 mL of the mobile phase (water and acetonitrile 90:10, v/v, with 0.2% formic acid) and analyzed.

## 2.8. Calculation of hazard quotient and risk estimation

The model of Hazard Quotient was utilized to estimate the risk of ingesting residues with milk. Hazard quotient is defined as the ratio of the prospective exposure to a material and the concentration where no harmful impacts are anticipated.

$$Hazard\ Quotient = \frac{Estimated\ Daily\ Intake\ (EDI)}{Acceptable\ Daily\ Intake\ (ADI)}$$

The estimated daily intake (EDI) was computed using the following equation presented by Juan et al. [19].

$$EDI = \frac{Concentration\ of\ Residue\ in\ \frac{\mu g}{kg} \times Daily\ Intake\ of\ milk\ in\ kg/person}{Adult\ Body\ Weight\ (kg)}$$

The mean level of residual antibiotics in raw milk was calculated. Then, the mean concentration and normal daily milk consumption based on a body weight of 60 kg and 10 kg for adults and children, respectively, were utilized for calculations. According to the data supplied by the Food and Agriculture Organization of the United Nations, the per capita availability of milk in Egypt was 96.98 mL/day [20].

Acceptable Daily Intake (ADI) is an approximated quantity of residue permitted to be consumed daily throughout a life expectancy with no noticeable health risk stated based on body weight. ADI of oxytetracycline is 0.03 mg/kg by/day [21].

If the hazard quotient is lower than or equal to one, this implies insignificant hazard whilst a value greater than one indicates harmful effects [22].

## 3. Results

### 3.1. Delvotest SP-NT (microbial inhibitor test)

In this study, the detection of antibiotic residues in milk samples was performed using a microbial inhibitor test and liquid chromatography-mass spectrometry. Results of Delvotest SP-NT revealed that 40 out of 200 tested samples showed no color change or partial color change, suggesting a positive result in 20% of the total samples as shown in (Table 2). Subsequently, all of the 40 positive samples were further tested using HPLC-MS/MS method as a confirmatory test.

**Table 2. Results of microbial inhibitor test (Delvotest SP-NT).**

| Sample Number | Result | Sample Number | Result |
|---------------|--------|---------------|--------|
| 1 | - | 117 | + |
| 2–4 | + | 118–133 | - |
| 5–17 | - | 134–140 | + |
| 18–27 | + | 141–167 | - |
| 28–52 | - | 168–176 | + |
| 53–58 | + | 177–192 | - |
| 59–80 | - | 193 | + |
| 81–83 | + | 194–200 | - |
| 84–116 | - | | |

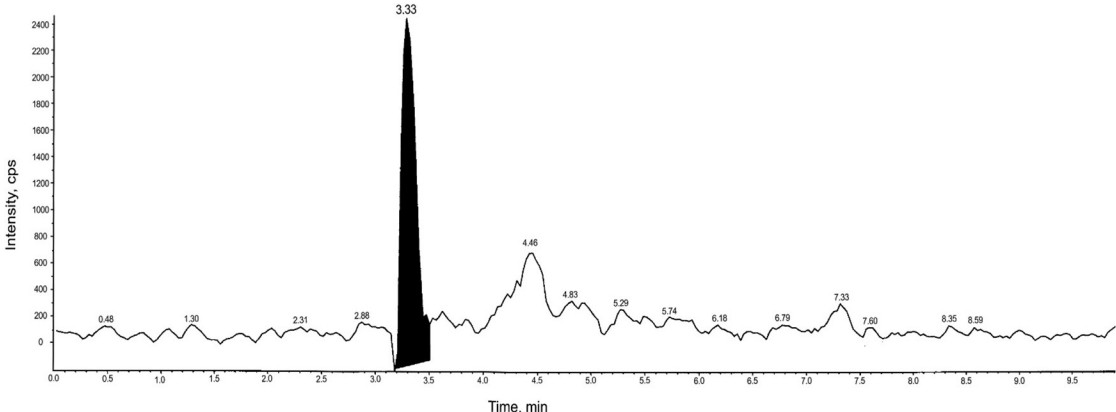

**Fig 1. Typical chromatogram of milk sample positive for oxytetracycline.**

### 3.2. HPLC-MS/MS

Results of HPLC-MS/MS revealed that only 4 out of 40 suspected samples analyzed (10%) were found to be truly positive for oxytetracycline residues. We could not detect any of the other tested antibiotics in the rest of the suspected positive samples via HPLC-MS/MS analysis which were consequently, considered false positives. The four positive milk samples originated farms in Egypt. The four positive samples contained 120, 132, 141, and 150 μg/kg of oxytetracycline residues, thus, exceeding the MRL which is 100 μg/kg [20]. It could be concluded that milk obtained from treated cows contained a residual antibiotic. The Chromatogram obtained because of the analysis is presented in (Fig 1).

### 3.3. Hazard quotient and risk estimation

The estimation of health risk for confirmed positive milk samples from dairy farms was conducted to verify possible threats to consumers resulting from the intake of milk containing antibiotic residues at levels exceeding the MRL. The mean oxytetracycline concentrations in the positive milk samples were 120, 132, 141, and 150 μg/kg. HQ for detected oxytetracycline remains in milk samples from dairy farms was computed to estimate any health threats to consumers (Table 3). The HQ for the detected residues of oxytetracycline in milk samples from

**Table 3. Estimation of human health risk based on hazard quotient for oxytetracycline residues through milk consumption in adults and children from dairy farms.**

| Mean Concentration (μg/kg) | Age Group | Body Weight[a] (kg) | EDI | ADI (3) | Hazard Quotient |
|---|---|---|---|---|---|
| 120 | Adults | 60 | 0.194 | 30 | 0.0065 |
| 120 | Children | 10 | 1.164 | 30 | 0.0388 |
| 132 | Adults | 60 | 0.213 | 30 | 0.0071 |
| 132 | Children | 10 | 1.280 | 30 | 0.0427 |
| 141 | Adults | 60 | 0.228 | 30 | 0.0076 |
| 141 | Children | 10 | 1.368 | 30 | 0.0456 |
| 150 | Adults | 60 | 0.242 | 30 | 0.0081 |
| 150 | Children | 10 | 1.455 | 30 | 0.0485 |

Abbreviations: ADI, acceptable daily intake; EDI, estimated daily intake.

[a]Body weights for different age groups were taken from FAO/WHO guidelines.

farms was less than one, which indicates insignificant adverse impacts on the consumer health as a result of the intake of the investigated samples.

## 4. Discussion

Low-cost testing methods are required for examining milk samples for the presence of antibiotic residues at levels above the levels set by community legislations. Due to their effectiveness, microbial inhibition methods have been largely used instead of physical-chemical methods. Those methods offer many advantages, such as their ease of use, no need for special training, simple equipment, and ability to detect a wide range of antibiotic residues within a single test [23]. The most commonly used tests are microbiological tests using *Bacillus stearothermophilus* spores, Delvotest SP, Copan Test, Charm Farm-960 Test, and others [16]. In this study, Delvotest SP-NT was used to detect antibiotic residues in milk samples collected from both the KSA and Egypt. Results of Delvotest SP-NT are observed visually as purple and yellow colors, which are easily recognized However, the samples containing intermediate concentrations of antibiotics that render the visual reading of the reaction more difficult [24, 25]. In those samples, the agar medium appeared as a mixture of purple color in a yellow background indicating a possible positive result. Moreover, visual estimation of results varies depending on the milk type and the mechanism of antibiotic action [25]. Therefore, microbiological testing is less appropriate for conclusive analyses leading to debatable results (false positives). Moreover, the presence of natural inhibitors in fresh milk may lead to false-positive Delvotest results as well [26]. These results should therefore be confirmed using more specific and sensitive techniques, such as HPLC-MS/MS. Based on the results of both Delvotest SP-NT and HPLC-MS/MS, it was concluded that positive samples were those collected from treated farm cows. This may be due to collection of milk just after antibiotic administration, drug misuse, or bad hygiene [27–30]. Delvotest results were similar to those reported by Hakem et al. [31] in Algeria, who detected no antibiotic residues in milk samples obtained from two Dairies Mitidja's Farms. In another study, about 10% of bulk tank milk samples and 20% of untreated bovine milk were reported positive [32]. These results were higher than those observed by Ben-Mahdi & Ouslimani in Algiers (9. 87%) [33]. On the other hand, other studies showed different results including, Zinedine et al. [27] in Morocco, Tarzaali et al. [34] in Mitidja, Aggad et al. [35] in the west of Algeria and Titouche et al. [36] in Tizi-Ouzu, where a higher percentage of antibiotic residues in milk ranging from (29–89%) was detected. HPLC-MS/MS revealed only four positive samples containing oxytetracycline residues as was reported in previous articles [37], and results were close to those reported by Martins et al. [38] in Brazil, who found 1.76% of antibiotic residues in milk samples. This low number may be attributed to the use of growth promoters rather than the use of antibiotics. Other studies produced a positive rate of more than 15% as reported by Li et al. [39] in China and García et al. [40] in Spain reporting 28%.

Health risk estimations were conducted for the confirmed positive milk samples and it was less than one, so it is presumed that there were insignificant adverse impacts on the consumer health associated with the intake of the investigated samples. Comparable results were stated by Moudgil et al. [41], who assessed the dietary exposure to residual antibiotics detected in raw and commercial milk samples in Punjab, India. The study stated no toxicological threat to consumers accompanying the intake of the examined milk samples concerning the antibiotics under study. Similar conclusions were also described by Rahman et al. [22], where the estimated dietary exposure to residual antibiotics through milk in Bangladesh was lower than the toxicological standard value.

## 5. Conclusion

One of the most significant concerns affecting public health is antibiotic residues found in milk. According to this study, Egyptian and Saudi Arabian cow's milk contained low levels of antibiotics. Ten percent of tested positive samples contained oxytetracycline residues exceeding the MRL after being examined with HPLC-MS/MS. These positive results were detected in samples obtained from the farms in Egypt. The occurrence of antibiotic residues in milk indicates the importance of further control of milk, which is tested using microbiological Delvotest SP-NT and confirmed with HPLC-MS/MS. We could conclude that HPLC-MS/MS could be considered a reliable analytical method for determining whether milk contains multiple antibiotics.

## Author Contributions

**Conceptualization:** Nora H. Al-Shaalan, Jenny Jeehan Nasr, Shereen Shalan.

**Data curation:** Jenny Jeehan Nasr, Shereen Shalan, Areej M. El-Mahdy.

**Formal analysis:** Jenny Jeehan Nasr, Shereen Shalan, Areej M. El-Mahdy.

**Funding acquisition:** Nora H. Al-Shaalan, Jenny Jeehan Nasr, Shereen Shalan, Areej M. El-Mahdy.

**Investigation:** Jenny Jeehan Nasr, Shereen Shalan, Areej M. El-Mahdy.

**Methodology:** Jenny Jeehan Nasr, Shereen Shalan, Areej M. El-Mahdy.

**Project administration:** Nora H. Al-Shaalan, Jenny Jeehan Nasr, Shereen Shalan, Areej M. El-Mahdy.

**Resources:** Nora H. Al-Shaalan, Jenny Jeehan Nasr, Shereen Shalan, Areej M. El-Mahdy.

**Software:** Jenny Jeehan Nasr, Shereen Shalan, Areej M. El-Mahdy.

**Supervision:** Jenny Jeehan Nasr, Shereen Shalan, Areej M. El-Mahdy.

**Validation:** Jenny Jeehan Nasr, Shereen Shalan, Areej M. El-Mahdy.

**Visualization:** Jenny Jeehan Nasr, Shereen Shalan, Areej M. El-Mahdy.

**Writing – original draft:** Jenny Jeehan Nasr, Shereen Shalan, Areej M. El-Mahdy.

**Writing – review & editing:** Jenny Jeehan Nasr, Shereen Shalan, Areej M. El-Mahdy.

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
