## [Decision Letter · Decision Letter 0]

16 Mar 2022

PONE-D-22-02795Inspection of Antimicrobial Remains in Bovine Milk in Egypt and Saudi Arabia Employing a bacteriological test kit and HPLC-MS/MS with Estimation of Risk to Human HealthPLOS ONE

Dear Dr. El-Mahdy,

Thank you for submitting your manuscript to PLOS ONE. After careful consideration, we feel that it has merit but does not fully meet PLOS ONE’s publication criteria as it currently stands. Therefore, we invite you to submit a revised version of the manuscript that addresses the points raised during the review process.

We look forward to receiving your revised manuscript.

Kind regards,

Joseph Banoub, Ph,D., D. Sc., FCIC. FRCS

Academic Editor

PLOS ONE

Journal Requirements:

Reviewers' comments:

Reviewer's Responses to Questions

**Comments to the Author**

1. Is the manuscript technically sound, and do the data support the conclusions?

Reviewer #1: Yes

Reviewer #2: Yes

2. Has the statistical analysis been performed appropriately and rigorously? 

Reviewer #1: N/A

Reviewer #2: Yes

3. Have the authors made all data underlying the findings in their manuscript fully available?

Reviewer #1: Yes

Reviewer #2: Yes

4. Is the manuscript presented in an intelligible fashion and written in standard English?

Reviewer #1: Yes

Reviewer #2: Yes

5. Review Comments to the Author

Reviewer #1: Dear Editor,

I have carefully read the manuscript “Inspection of Antimicrobial Remains in Bovine Milk in Egypt and Saudi Arabia Employing a bacteriological test kit and HPLC-MS/MS with Estimation of Risk to Human Health”. The manuscript represents the application of bacteriological and HPLC-MS/MS analyses on milk samples from Egypt and Saudi Arabia in order to inspect antimicrobial remains and estimate the health risk of their presence in bovine milk.

I think that the manuscript needs to be re-edited; the present form of paper is not acceptable for publication. Accordingly, after mandatory major revisions, it can be published in your journal.

Having said that I would like to make a few observations and offer a few comments:

1. Lines 106-112: The Literary review written in the Discussion should appear in this paragraph, in the Introduction (see comment 6 below). The novelty of the research should be compared to the other existing studies- what are the differences of this study from previous studies? If there are any...

2. Line 122: 5 different Governorates (Cairo, Giza, Dakahlia, Gharbia, Sharkia, Kafr El Sheikh, and Qalyoubia) - these are 7 Governorates.

3. Line 138: fermented milk? What’s this??

4. Lines 247-248: “The Chromatogram obtained because of the analysis is presented in fig. 2” - needs to be edited (bad language)

5. The Discussion section must be re-organized and re-written.

6. The literary review is not appropriate for the Discussion. It should appear in the Introduction and can be addressed in the discussion. This review doesn’t contribute to the discussion and only interferes with the understanding of the researchers' arguments.

7. Line 277: “Non-specificity is the main limitation of these microbial assays”- written in an inadequate place in the paragraph.

8. Lines 282-3: “Moreover, different milk types and different antibiotic mechanisms of action make it more challenging to assess due to their different colors” – this sentence doesn’t make sense, explain the end of the sentence.

9. Line 286: incomplete milking? Freshly cows? What do these phrases mean??

10. Line 302: “Those detected a higher percentage of antibiotic residues in milk (57, 89, 29, and 46 %, respectively).” The percentages refer to the antibiotic residues in milk? Or rather to samples that contained antibiotic residues? It seems more reasonable that the second option is correct...

11. Lines 303-306: same comment as in line 302.

12. Lines 307-316: repetitive – written already in sections 2.8 and 3.3. These lines should be summarized.

13. Lines 324-336: the conclusion section should be edited. This section should be concise and there is no need to repeat the results that appear already in the Results and Discussion sections. Instead, the advantage of the methods proposed in the manuscript should be highlighted.

14. Figure 2: the X axis title is missing. The authors presented the HPLC chromatogram alone. They would improve the graphic quality of the manuscript if they would add the MS spectrum of the sample or a HPLC chromatogram of an oxytetracycline standard sample in order to confirm the identification of oxytetracycline.

15. Supporting information: all figures and tables are presented in the manuscript and therefore, don’t need to be added to the SI as well (-erase them from the SI). The SI should contain all the results that haven’t been presented in the manuscript, e.g.: the negative results of the Microbial inhibitor test (Delvotest SP-NT), the HPLC-MS chromatograms of all, or at least the three other positive samples and one negative sample, analyzed by this method.

Reviewer #2: Thanks for your valuable work where in this manuscript authors proved that milks from KSA and Egypt considered free from antibiotics as only 10% of all suspicious sample contained antibiotics. But some changes must be done as mentioned in the attached file

6. PLOS authors have the option to publish the peer review history of their article (what does this mean?). If published, this will include your full peer review and any attached files.

Reviewer #1: No

Reviewer #2: **Yes: **Doaa A. Ghareeb

---

## [Author Response · Author response to Decision Letter 0]

2 Apr 2022

Dear editor: 

Thanks for your suggestions. They were very helpful. They were all incorporated in my revision.

• Figure files were added to the Preflight Analysis and Conversion Engine (PACE) digital diagnostic tool.

• For the minimal data set underlying the results, all relevant data are provided in the manuscript.

Dear Reviewer 1: 

I have incorporated all of your suggestions into my revision. They were very helpful. Thank you.

1- The manuscript was re-edited by Dr. Hanan Shehata, Postdoctoral fellow, University of Guelph.

2- All your valuable comments were done and it will be in the revised manuscript with track changes.

Dear Reviewer 2: 

• Thank you for your valuable comments.

• All of your comments were done and it will be in the revised manuscript with track changes.

---

## [Editor Report · Decision Letter 1]

14 Apr 2022

Inspection of Antimicrobial Remains in Bovine Milk in Egypt and Saudi Arabia Employing a bacteriological test kit and HPLC-MS/MS with Estimation of Risk to Human Health

PONE-D-22-02795R1

Dear Dr. El-Mahdy,

We’re pleased to inform you that your manuscript has been judged scientifically suitable for publication and will be formally accepted for publication once it meets all outstanding technical requirements.

Kind regards,

Joseph Banoub, Ph,D., D. Sc., FCIC, FRCS

Academic Editor

PLOS ONE
---

## [Editor Report · Acceptance letter]

19 Apr 2022

PONE-D-22-02795R1 

Inspection of Antimicrobial Remains in Bovine Milk in Egypt and Saudi Arabia Employing a bacteriological test kit and HPLC-MS/MS with Estimation of Risk to Human Health 

Dear Dr. El-Mahdy:

I'm pleased to inform you that your manuscript has been deemed suitable for publication in PLOS ONE. Congratulations! Your manuscript is now with our production department. 

Kind regards, 

on behalf of

Dr. Joseph Banoub 

Academic Editor

PLOS ONE